# Synthesis, Antifungal Activity, Cytotoxicity and QSAR Study of Camphor Derivatives

**DOI:** 10.3390/jof8080762

**Published:** 2022-07-22

**Authors:** Xinying Duan, Li Zhang, Hongyan Si, Jie Song, Peng Wang, Shangxing Chen, Hai Luo, Xiaoping Rao, Zongde Wang, Shengliang Liao

**Affiliations:** 1East China Woody Fragrance and Flavor Engineering Research Center of National Forestry and Grassland Administration, Camphor Engineering Research Center of National Forestry and Grassland Administration, College of Forestry, Jiangxi Agricultural University, Nanchang 330045, China; duan19990226@163.com (X.D.); zhangli_miaomuli@163.com (L.Z.); hongyansi16@163.com (H.S.); pengwang1981@126.com (P.W.); csxing@126.com (S.C.); luohai87@126.com (H.L.); 2Department of Natural Sciences, University of Michigan-Flint, 303E Kearsley, Flint, MI 48502, USA; jiesong@umich.edu; 3College of Chemical Engineering, Huaqiao University, Xiamen 361021, China; rxping2001@163.com

**Keywords:** camphor, synthesis, antifungal activity, cytotoxicity, QSAR

## Abstract

Control of fungal phytopathogens affecting crops and woodlands is an important goal in environmental management and the maintenance of food security. This work describes the synthesis of 37 camphor derivatives, of which 27 were new compounds. Their antifungal effects on six fungi were evaluated in vitro. Compounds **3a**, **4a** and **5k** showed strong antifungal activity against *Trametes versicolor*, with EC_50_ values of 0.43, 6.80 and 4.86 mg/L, respectively, which were better than that of tricyclazole (EC_50_ 118.20 mg/L) and close to or better than that of carbendazim (EC_50_ 1.20 mg/L). The most potent compound, **3a**, exhibited broad-spectrum antifungal activity towards six fungi with EC_50_ values within the range of 0.43–40.18 mg/L. Scanning electron microscopy demonstrated that compounds **3a**, **4a** and **5k** gave irregular growth and shriveling of the mycelia. In vitro cytotoxicity evaluation revealed that the tested camphor derivatives had mild or no cytotoxicity for LO_2_ and HEK293T cell lines. Quantitative structure−activity relationship (QSAR) analysis revealed that the number of F atoms, relative molecular weight, the atomic orbital electronic population and total charge on the positively charged surfaces of the molecules of camphor derivatives have effects on antifungal activity. The present study may provide a theoretical basis for a high-value use of camphor and could be helpful for the development of novel potential antifungals.

## 1. Introduction

In modern agriculture, phytopathogenic fungi not only severely reduce grain yield and quality, but also secrete toxins that endanger the health of humans and animals [1]. In addition, wood rot fungi can seriously degrade wood products [2]. At present, the chemical fungicides widely used to control phytopathogenic and wood rot fungi are economical, easy to use and efficient. However, the misuse of chemical fungicides can result in the rapid development of resistance and serious environmental contamination [3,4]. The development of novel highly efficient, low-toxicity and ecofriendly alternative fungicides is urgently needed. Preparation of novel antifungals from plant essential oils and some of their main components has attracted attention because of their broad-spectrum antifungal properties, mechanisms of action and environmental compatibility [5,6,7,8,9,10,11,12].

The naturally occurring monoterpenketone camphor is found in *Cinnamomum camphor*, and it can also be obtained by the isomerization–oxidation of turpentine. Due to its biological activity and variable molecular structure, camphor has potential as a building block in the preparation of biologically active derivatives [13,14,15,16]. The development of camphor-based anti-microbial derivatives has recently been reported [17,18,19,20,21,22]. For example, Ma et al. [21] synthesized novel camphoric-acid-based acylhydrazone derivatives, with some of the most active compounds giving growth inhibition of 90.00%-95.00% at 0.05 mg/mL. Such results suggest the feasibility of developing novel anti-microbial agents with camphor as a raw material.

Thiosemicarbazides (TSCs) have attracted much attention because they can impart antifungal properties on compounds [23,24]. For instance, Qin et al. [25] synthesized novel thiosemicarbazone chitosan derivatives and demonstrated these compounds exhibited broad-spectrum antifungal activity against four common crop-threatening pathogenic fungi. Furthermore, Yamaguchi et al. found that TSC–camphene derivatives had excellent antifungal effects on *Trichophyton menthagophytes* through damage to the fungal cell wall. The MIC value of the most active antifungal compound was 55.00 μmol L^−1^ [26]. The potential antifungal activity of camphor and TSCs focused our efforts on the development of some novel TSC–camphor derivatives as antifungal candidates.

Herein, two series of camphor derivatives have been designed, synthesized and characterized. Their antifungal effects on *Phytophthora nicotianae*, *Fusarium verticillioides*, *Colletotrichum gloeosporioides*, *Sphaeropsis sapinea*, *Fusarium oxysporum* and *Trametes versicolor* were assessed. The in vitro cytotoxicity of representative camphor derivatives were evaluated on LO_2_ and HEK293T. The morphological changes in camphor derivatives against *T. versicolor* were assessed using scanning electron microscopy (SEM). Quantitative structure–activity relationship (QSAR) analysis was used to detect the relationships between structural features and antifungal effects on *T. versicolor*. The current study may provide a theoretical basis for the high-value utilization of camphor, and could help the development of novel potential antifungals against phytopathogenic and wood rot fungi.

## 2. Materials and Methods

### 2.1. Materials and Equipment

Camphor, thiosemicarbazide, bromobenzyl derivatives, *α*-bromoacetophenone derivatives, carbendazim and tricyclazole were purchased from Shanghai Aladdin Reagent Co., Ltd. (Shanghai, China). Ethanol, isopropanol, acetonitrile, petroleum ether, dimethyl sulfoxide and ethyl acetate were purchased from Xilong Science Co., Ltd. (Guangdong, China). Potato dextrose agar (PDA) were purchased from Huan Kai Microbial Technology Co., Ltd. (Guangdong, China). Melting points were determined with the WRS-2 melting point apparatus (Shanghai Precision and Scientific Instrument Co., Ltd., Shanghai, China) and were uncorrected. FT-IR analysis was performed on the Nicolet iS50 Fourier Transform Infrared Spectrometer (Thermo, Waltham, MA, USA). The ^1^H NMR and ^13^C NMR spectra were measured on an AVANCE 400 spectrometer (Thermo, Waltham, MA, USA) using CDCl_3_ or DMSO-*d*_6_ as the solvent. High-resolution mass spectrometry (HRMS) spectra were recorded on a triple time of flight TOF 5600+ (AB Sciex) mass spectrometer (Concord, ON, Canada). Cytotoxicity data was measured by microplate reader Tecan Infinite 200 Pro (Tecan Trading AG, Maennedorf, Switzerland) at 492 nm. The mycelium morphologies of *T. versicolor* were observed using a scanning electron microscope (Hitachi, Regulus 8100, Tokyo, Japan).

Five phytopathogenic fungi, namely *Phytophthora nicotianae* (*P. nicotianae*), *Fusarium verticillioides* (*F. verticillioides*), *Colletotrichum gloeosporioides* (*C. gloeosporioides*), *Sphaeropsis sapinea* (*S. sapinea*) and *Fusarium oxysporum* (*F.*
*oxysporum*) were isolated and identified by the plant pathology laboratory at the College of Agriculture, Jiangxi Agricultural University. The wood rot fungus *Trametes versicolor* cfcc5336 was purchased from China Forestry Culture Collection Center. All strains and incubated in PDA at 26 °C.

### 2.2. Synthesis of Camphor Derivatives

#### 2.2.1. Synthesis of 2-(1,7,7-Trimethylbicyclo [2.2.1]heptan-2-ylidene)hydrazine-1-carbothioamide (**3a**)

Camphor (10 mmol) and thiosemicarbazide (10 mmol) were dissolved in a round-bottom flask (50 mL) containing isopropanol (20 mL) and 10% HCl (1 mL). The mixture was stirred for 4 h at room temperature. The progress of the reaction was checked by thin-layer chromatography (TLC) with the developing solvent ethyl acetate/petroleum ether (4.5:1, V/V). The precipitates were collected by vacuum filtration in a fume hood, and the filtered cake was washed with petroleum ether. The solid powder was recrystallized with ethanol and H_2_O, filtered under vacuum, washed with petroleum ether (3 × 100 mL) and dried in an oven to obtain the white solid compound **3a**. The synthetic routes to compound **3a** and its derivatives are shown in Figure 1. Compound **3a** was characterized by ^1^H NMR, ^13^C NMR, FT-IR and HRMS (Appendix A).

#### 2.2.2. Synthesis of N-(2-Chlorobenzyl)-2-(1,7,7-trimethylbicyclo[2.2.1]heptan-2-ylidene)hydrazine-1-carbothioamide compounds (**4a**–**4s**)

Substituted benzyl bromide (12 mmol) was added dropwise over 30 min to a stirred solution of compound **3a** (10 mmol) in acetonitrile (20 mL) that was stirred and heated to 80 °C, for 5 h. After the reaction was complete, solids precipitated on the bottom of the round-bottom flask. The precipitates were collected through vacuum filtration in a fume hood, and the filtered cake was washed with petroleum ether. The solid powder was recrystallized with ethanol and H_2_O, filtered under vacuum, washed with petroleum ether (3 × 100 mL) and dried in an oven to give the pure product (**4a**–**4s**). The synthetic routes to compound **4a**–**4s** are shown in Figure 1. Compound **4a**–**4s** were characterized by ^1^H NMR, ^13^C NMR, FT-IR and HRMS (Appendix A).

#### 2.2.3. Synthesis of 4-Phenyl-2-(2-(1,7,7-trimethylbicyclo[2.2.1]heptan-2-ylidene)hydrazinyl)thiazole compounds (**5a**–**5o**)

A stirred solution of compound **3a** (10 mmol) in ethanol (50 mL) was added to *α*-bromoacetophenone derivatives (10 mmol) in a round-bottom flask (100 mL). The mixture was stirred and heated to 85 °C, for approximately 3 h, evaporated under reduced pressure to remove the organic phase, and the solid powder was collected by vacuum filtration in a fume hood. Then, the solid powder was recrystallized with ethanol, filtered under vacuum, washed with petroleum ether (3×100 mL) and dried in an oven to give the pure product (**5a**–**5o**). The synthetic routes to compound **5a**–**5o** are shown in Figure 1. Compound **5a**–**5o** were characterized by ^1^H NMR, ^13^C NMR, FT-IR and HRMS (Appendix A).

### 2.3. In Vitro Antifungal Activity Evaluation

The in vitro antifungal activity of the camphor derivatives against six fungi was evaluated using the mycelium growth rate method [27], with the commercial fungicides tricyclazole and carbendazim serving as positive controls. Dimethyl sulfoxide was used as a carrier control. The camphor derivatives dissolved in dimethyl sulfoxide to generate stock solutions (1.0 × 10^4^ mg/L), which were further diluted to 250, 125, 62.5, 31.3, 15.6, 7.81, 3.91, 1.95, 0.977, 0.488 and 0.244 mg/L, respectively, in PDA plates. Mycelial disks (0.50 cm) of activated fungi were added to PDA plates and incubated at 26 °C for 4–7 days. All tests were repeated three separate experiments. The Cross method was used to measure the colony diameter in the Petri dishes of treatment groups, once the colony diameter reached approximately 68–72 mm in the Petri dish of the untreated control. The inhibition rate (IR) reflected the antifungal activity of each compound and was calculated as Equation (1):IR(%) = (B − T)/(B − 0.5) (cm) × 100%(1)
where B and T are the diameters of the blank control and treatment groups, respectively. The effective concentration giving 50% inhibition (EC_50_) for each of the tested compounds was analyzed by probit analysis within the SPSS statistics software.

### 2.4. Cytotoxicity (MTT) Assays

The 3-(4, 5-dimethylthiazolyl-2)-2,5-diphenyltetrazolium bromide (MTT) assay was used to assess the in vitro cytotoxicity of the compounds on two human cell lines, LO_2_ and HEK293T [28]. The LO_2_ and HEK293T were cultured in Dulbecco’s modified eagle medium (DMEM). In the tested group, the cell lines were treated with 10 mg/L of the compounds for 24 h in 96-well plates, and cell lines treated without compounds were used as the blank control. After 24 h treatment, 20 μL MTT solution (5 mg/mL) was added to each well, followed by 4 h incubation. Finally, the absorbance of the wells at 492 nm was measured using a microplate reader Tecan Infinite 200 Pro. The experiment was repeated three times. Cell viability was evaluated as the ratio of the absorbance of antifungal treated to untreated cells.

### 2.5. Scanning Electron Microscope (SEM) Observations

Freshly cultured mycelia were inoculated on PDA plates containing no compound, compounds **3a**, **4a**, or **5k** and incubated at 26 °C, for 7 days, placed on a glass slide and dehydrated [29]. The morphology of the mycelia was observed by SEM.

### 2.6. Quantitative Structure−Activity Relationship (QSAR) Study

The structure of each of the synthesized compounds was drawn using Gauss view 5.0.9 software (Gaussian, Inc. Wallingford, CT, USA). The density functional theory (DFT) method at the B3LYP 6-31G(d) level of Gaussian 09W (Gaussian, Inc. Wallingford, CT, USA) [30] was used to optimize the molecular structures and calculate the minimum energy of the camphor derivatives. Then, Ampac 8.16 software (Semichem, Inc. Shawnee, KS, USA) [31] was used to convert the Gaussian 09W output files into Ampac output files, which were compatible with Codessa 2.7.10 software (Semichem, Inc. Shawnee, KS, USA) [32]. Twenty-nine camphor derivatives were selected and assigned to the modeling dataset, and the remaining camphor derivatives (compounds **4q**, **4i**, **4j**, **4l**, **5i**) were randomly selected for assignment to the external validation dataset. Log_50_EC_50_ of the camphor derivatives was taken as the dependent variable of the QSAR model equation, while the molecular structure descriptors of camphor derivatives provided the independent variables of the QSAR model. Regression analysis of the molecular structures and antifungal activity was carried out using the heuristic method encoded in Codessa 2.7.10 software [33], and the quantitative relationship models were constructed. The best QSAR model was determined by the “breaking point” criterion (ΔR^2^<0.02–0.04) [34]. Finally, “leave-one-out” cross-validation, internal validation and external validation were used to validate the QSAR model.

## 3. Results and Discussion

### 3.1. Chemistry

The synthesis of **3a**, **5a**, **5d**, **5f**, **5g**, **5i**, **5k** and **5l** was first reported by Sokolova et al. [35]. Here, two series of 35 camphor derivatives were synthesized, of which 27 were new compounds. In Sokolova’s study [35], via catalysis using sulfuric acid, compound **3a** was prepared by reacting camphor with thiosemicarbazide for 10 h, with a yield of 74.00%. In the present study, hydrochloric acid was used as catalyst instead of sulfuric acid, the reaction time was shortened to 4 h, and the yield of compound **3a** was increased to 93.24 % (Appendix A).

Compounds **4a**–**4s** were produced via the reaction between compound **3a** and benzyl bromide or substituted benzyl bromide, at 80 °C, for 5 h (Figure 1). The progress of the reaction was monitored by TLC. All the products were purified by recrystallization. The yields of compounds **4a**–**4s** reached 78.15~87.22% (Appendix A). This synthetic route had the advantage of a short reaction time, fewer by-products and simple post-treatment method.

Compounds **5a**–**5o** were synthesized via the reaction between compound **3a** and α-bromoacetophenone or substituted α-bromoacetophenone, at 85 °C, for 3 h (Figure 1), with ethanol used as a solvent. The yields of the products reached 49.98~92.11% (Appendix A). In Sokolova’s study [35], N-Bromosuccinimide and p-toluenesulfonic acid were used as catalysts, with a mixture of ethanol and CHCl_3_ used as a solvent; the reaction was carried out for 24 h, and the product yield was 34.00~71.00%. In the present work, the reaction conditions were optimized, no catalyst was used, the solvent was simplified to ethanol, the reaction temperature was increased to 85 °C, and the products were more efficiently prepared, with ideal yields. The products were characterized by ^1^H NMR, ^13^C NMR, FT-IR and HRMS. The structural characterization data indicated that compounds **3a**, **4a–4s** and **5a–5o** were successfully synthesized (Appendix A).

### 3.2. In Vitro Antifungal Activity and Structure−Activity Relationship (SAR)

The antifungal activity of camphor derivatives was tested against *P. nicotianae*, *F. verticillioides*, *C. gloeosporioides*, *S. sapinea*, *F. oxysporum* and *T. versicolor*. The commercial fungicides tricyclazole and carbendazim were used as positive controls. The results showed the camphor derivatives exhibited antifungal activity against six fungi to various extents (Figure 2, Table 1, Appendix A). *T. versicolor* was the most sensitive organism to the camphor derivatives compared with the other five fungi. Camphor and thiosemicarbazone had a weak inhibitory effect on six fungi. Compounds **3a** and **4a**–**4s** displayed low to medium activity, with EC_50_ values ranging from 25.08 to >1000 mg/mL against *P. nicotianae*, 31.32 to >1000 mg/mL against *F. verticillioides*, 12.85 to >1000 mg/mL against *C. gloeosporioides*, 17.09 to >1000 mg/mL against *S. sapinea* and 15.89 to >1000 mg/mL against *F. oxysporum*. Compounds **5a**–**5o** showed almost no antifungal activity against these five fungi.

However, the camphor derivatives (**3a**, **4a**–**4s** and **5a**–**5o**) exhibited moderate to potent antifungal activity against *T. versicolor*. Of these compounds **3a**, **4a**, **4d**, **4o**, **5a**, **5g**, **5k** and **5l** showed potent antifungal activity against *T. versicolor*, with EC_50_ values of 0.43, 6.80, 7.67, 6.89, 4.42, 7.85, 4.86 and 5.09 mg/L, respectively, which were superior to the values obtained for tricyclazole (EC_50_ 118.20 mg/L, Appendix A) and comparable to or better than those for carbendazim (EC_50_ 1.20 mg/L, Appendix A). At 15.6 mg/L, the inhibition by compounds **3a**, **4a**, **4d**, **4o**, **5a**, **5g**, **5k** and **5l** against *T. versicolor* was >50.00% (Appendix A). Notably, compound **3a** displayed broad-spectrum antifungal activity; its EC_50_ values against six fungi were 25.10, 40.18, 12.85, 17.09, 19.30 and 0.43 mg/L, respectively (Table 1 and Figure 2). It exhibited a stronger antifungal effect on six fungi than the positive control tricyclazole (EC_50_: 80.58, 185.93, 79.39, 268.37, 66.78 and 118.20 mg/L, respectively). Even at 0.244 mg/L, the inhibition by compound **3a** against *T. versicolor* was >33.20% (Figure 2, Appendix A).

*T. versicolor* is a common polypore species found throughout the world. It grows easily under different types of environment and is a suitable model organism for screening new antifungal agents. In the present study, camphor derivatives had good antifungal activity against *T. versicolor* and more modest activity against another five phytopathogenic fungi, indicating that camphor derivatives could serve as leading compounds in the discovery of novel broad-spectrum antifungal agents.

Previous studies have reported that thiosemicarbazide (TSC) is a multi-target compound that can affect the cell cycle regulation of fungi and inhibit the metabolism of various nutrients [36]. Hence, it has been widely used in the preparation of fungicides, such as Thiophanate-Methyl [37]. In the present study, the thiosemicarbazide group was fused with the camphor backbone, in order to obtain derivatives with good activity against fungi. As anticipated, compound **3a**, the product formed by the reaction of thiosemicarbazide and camphor, exhibited much better antifungal activity than thiosemicarbazide and camphor. This result indicated that fusion of the thiosemicarbazide group and the camphor skeleton greatly enhanced the antifungal activity of the derivative.

Some broad conclusions can be drawn from analysis of structure–activity relationships of camphor derivatives against *T. versicolor*. Most of the compounds, namely, **4a**–**4s** and **5a**–**5o**, exhibited better antifungal activity against *T. versicolor* compared with camphor and thiosemicarbazone. A comparison of the effects of the substituents on the benzene ring on antifungal activity against *T. versicolor* showed compounds **4a** (EC_50_ 6.80 mg/L) and **5a** (EC_50_ 4.42 mg/L), with no substituents in the benzene ring, displayed better antifungal activity than the other derivatives **4****b**–**4s** and **5****b**–**5o**. The introduction of additional electron-withdrawing or electron-donating groups on the benzene ring did not enhance the antifungal activity of the derivatives against *T. versicolor*. When the same substituent groups were introduced into the benzene ring of the derivatives, the number of substituent groups had improved the antifungal activity against *T. versicolor* [38]. It was found that derivatives monosubstituted with fluorine exerted stronger antifungal activity than those polysubstituted with fluorine (**4b–4d** > **4l–4n**; **5b–5d** > **5m**) corroborating of prior research [39]. The analysis of the effect of substituent position in the benzene ring on antifungal activity against *T. versicolor* showed that derivatives with a substituent group at the para-position generally gave better antifungal activity (**4d** > **4b**, **4c**; **4k** > **4i**, **4j**; **5k** > **5j**, etc.).

### 3.3. Evaluation of Cytotoxicity

Fourteen camphor derivatives (**1**, **3a**, **4a**, **4d**, **4p**, **5a–5d**, **5g**, **5h** and **5j**–**5l**) with good antifungal activity against *T. versicolor* were evaluate for cytotoxicity on two human cell lines, LO_2_ and HEK293T, using the MTT assay (Figure 3). All the tested compounds exhibited mild to no cytotoxicity for LO_2_ and HEK293T cells at a concentration of 10 mg/L. Figure 3c,d show that the LO_2_ and HEK293T cells treated by the tested derivatives remained healthy and round. Among the tested compounds, the raw material camphor was found to be non-toxic for LO_2_ and HEK293T cells. The intermediate compound 3a, with the best antifungal activity against *T. versicolor,* showed mild cytotoxicity towards LO_2_ and HEK293T cells (proportional cell viability> 90% at 10 mg/L, Figure 3a,b). The other camphor derivatives were not toxic towards LO_2_ and HEK293T cells.

LO_2_ and HEK293T represent cells normally found in human liver and kidney, respectively, where they are important cells for metabolism. Since pesticides may be ingested during use, it was important to evaluate the toxicity of pesticides on such cells. Our results show the camphor derivatives synthesized in present study have low toxicity or are non-toxic for LO_2_ and HEK293T cells. These results suggest that these camphor derivatives may be promising lead compounds in the development of low-toxicity antifungals.

### 3.4. Effect on the Mycelium Morphology of T. versicolor

Compounds **3a**, **4a** and **5k**, with excellent antifungal activity against *T. versicolor*, were evaluated for their effects on the morphology of *T. versicolor*. The ultrastructural of *T. versicolor* mycelia treated with compounds **3a**, **4a** and **5k** and a reagent blank control were observed by SEM. The results in Figure 4 and Appendix A show that the surface of *T. versicolor* mycelium treated with compounds **3a**, **4a** and **5k** become folded and shriveled (Figure 4b–d). In contrast, the hyphae of the untreated control group showed a smooth surface and normal growth (Figure 4a). Together with previous research findings [40,41], we speculate that the fungal cell walls and cell membranes of *T. versicolor* are potential targets of the camphor derivatives, but the mechanism of action of camphor derivatives needs further study.

### 3.5. QSAR Study

A QSAR analysis was carried out to identify quantitative relationship between the molecular structures of the camphor derivatives and their antifungal activity against *T. versicolor*. A series of QSAR models with different numbers of descriptors were obtained using the heuristic method encoded in Codessa 2.7.10. According to the “breaking point” criterion, the best QSAR model, which contained four descriptors (R^2^ = 0.911, F = 61.5, s^2^ = 0.0973), was obtained through linear regression analysis (Figure 5a). The details of the model are listed in Table 2, and a comparison of experimental and calculated log_50_ EC_50_ values is shown in Figure 5b. The figure shows a strong correlation resulting in only small differences between the experimental log_50_ EC_50_ values and the calculated log_50_ EC_50_ values, thus indicating that the best QSAR model is satisfactory.

In order to verify the robustness of the best QSAR model, “leave-one-out” cross-validation and internal validation were carried out, and the results are shown in Table 3. There was an acceptable difference between the squared correlation coefficients R^2^ (0.911), Rloo^2^ (0.844), R_training_^2^ (0.894) and R_test_^2^ (0.880) of the best QSAR model, indicating that the QSAR model was robust. In addition, the predictability of the best QSAR model was validated by external validation, and the results are illustrated in Figure 6. The squared correlation coefficient of external validation R_external_^2^ was 0.654, confirming that the prediction ability of the best QSAR model was satisfactory.

Based on the data listed in Table 2, the best QSAR model can be written as a linear regression equation:log_50_ EC_50_ = 1.757 + 0.416 × NO + 0.903 × RMW − 11.104 × MAOEP + 0.528 × FPSA-2N = 29, R^2^ = 0.911, F = 61.5, s^2^ = 0.0973(2)

N represents the number of camphor derivatives in the modeling dataset, R^2^ represents the squared value of the correlation coefficient of the model, F represents the Fisher test value, and s^2^ represents the variance of the model. Moreover, because the result of the t-test was positively correlated with the importance of the descriptors, it could be determined that the *Relative molecular weight* and *Number of F atoms* were the two most important descriptors affecting the antifungal activity of the camphor derivatives against *T. versicolor*.

According to Codessa’s Reference Manual [42,43], the descriptors used to construct the best QSAR model can be explained as follows.

The first descriptor was *Number of F atoms*, which is classified as a constitutional descriptor. This descriptor was positively related to the log_50_ EC_50_ value of the camphor derivatives, i.e., it was negatively related to the antifungal activity of the camphor derivatives against *T. versicolor*. A previous study had shown that the introduction of F atoms could increase drug permeability across the cell membrane and allow small-molecule compounds to penetrate the fungal cell membrane more easily, a feature that can strongly affect the antifungal activity of compounds [44]. Based on the SAR and QSAR of the present study, although fluorine atoms improved the antifungal activity of the camphor derivatives, greater numbers of fluorine atoms did not necessarily increase the antifungal activity of the camphor derivatives. Interestingly, a previous study drew similar conclusions [39].

The second most important descriptor was *Relative molecular weight*, which is also a constitutional descriptor. This descriptor was negatively related to the antifungal activity of the camphor derivatives. Generally, a low molecular weight most effectively improve the efficient binding between target proteins with compound receptors [45]. Our in vitro antifungal activity study found that the compounds **3a**, **4a** and **5a**, with lower molecular mass, showed better antifungal activity. These results suggest that molecular weight is an important parameter to be considered when designing novel camphor derivatives.

The third descriptor, *Minimum atomic orbital electronic population*, belongs to the family of quantum-chemical descriptors. It is a charge-distribution-related descriptor connected with the orbital electronic population [33]. It is a vital physical parameter that describes the nucleophilicity of compounds [46] that could affect the interaction between camphor derivatives and their targets. In the best QSAR model, this descriptor was the sole descriptor with a positive contribution to the antifungal activity of the camphor derivatives. We speculated that a lower value for minimum atomic orbital electronic population helped improve the antifungal activity of the camphor derivatives.

The fourth descriptor was *FPSA-2 Fractional PPSA (PPSA-2/TMSA) [Quantum-Chemical PC]*, which is an electrostatic descriptor [47]. This descriptor describes the total number of charges contained on the positively charged surface of a molecule and is closely related to molecular charge distribution and shape. This descriptor negatively correlated with the antifungal activity of the camphor derivatives. It showed that a lower total charge number on the positively charged surface of a molecule improved the antifungal activity of camphor derivatives.

In summary, the QSAR analysis revealed that fewer fluorine atoms, a lower relative molecular weight, a lower value for minimum atomic orbital electronic population, and a lower total charge on the positively charged surface of the molecule helped improve the antifungal activity of the camphor derivatives against *T. versicolor*.

## 4. Conclusions

Two series of camphor derivatives were successfully and efficiently synthesized in this work. Bioassays showed that the camphor derivatives **3a**, **4a** and **5k** showed potent antifungal activity against *T. versicolor* and low toxicity against two human cell lines, LO_2_ and HEK293T, suggesting that they may be promising lead compounds for the development of low-toxicity antifungals. SEM analysis showed that the camphor derivatives can markedly change the mycelial morphology of *T. versicolor*. In addition, SAR and QSAR studies provide useful information for the further design of camphor derivatives with improved antifungal activity. These findings lay a foundation and provided a reference for the development of camphor derivatives as novel antifungals. However, the mechanism of action of camphor derivatives has yet to be determined, and research on this aspect will be carried out in a further study.

## Figures and Tables

**Figure 1 jof-08-00762-f001:**
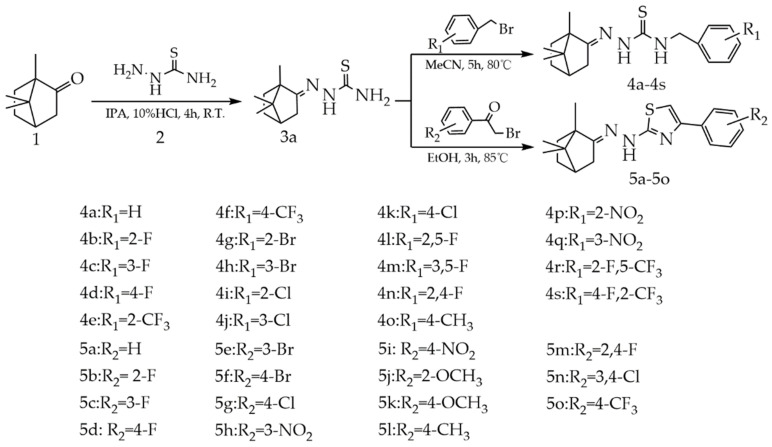
Synthetic routes of camphor derivatives.

**Figure 2 jof-08-00762-f002:**
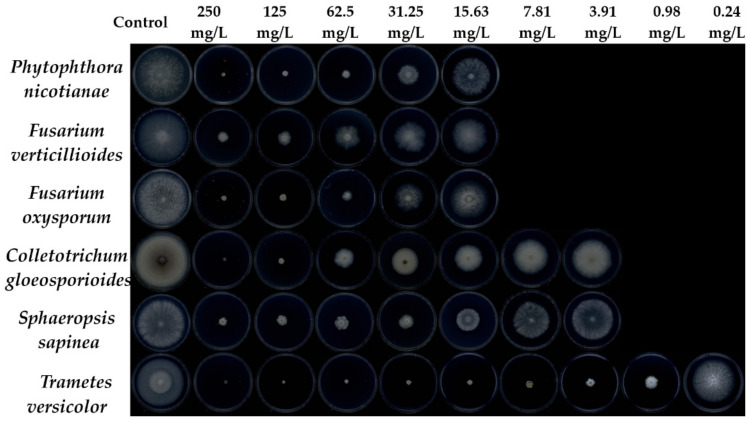
Photographs of antifungal experiments of compound **3a** against six fungi.

**Figure 3 jof-08-00762-f003:**
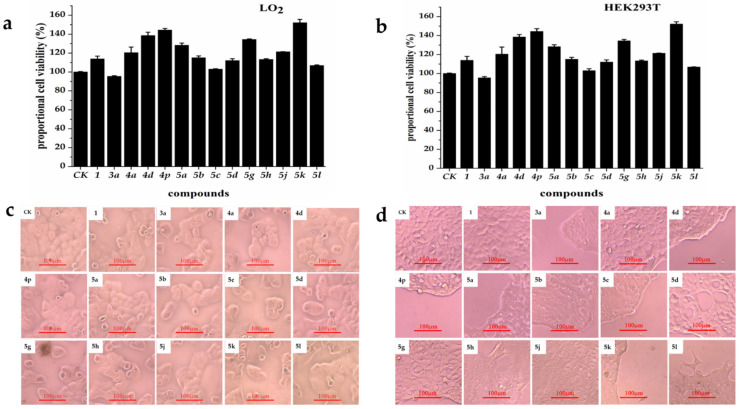
In vitro cytotoxicity of several camphor derivatives: (**a**) proportional viability of LO_2_ cells in response to 10 mg/L of compound; (**b**) proportional viability of HEK293T cells in response to 10 mg/L of compound; (**c**) photographic image of LO_2_ cells in response to 10 mg/L of compound; (**d**) photographic image of HEK293T cells in response to 10 mg/L of compound.

**Figure 4 jof-08-00762-f004:**
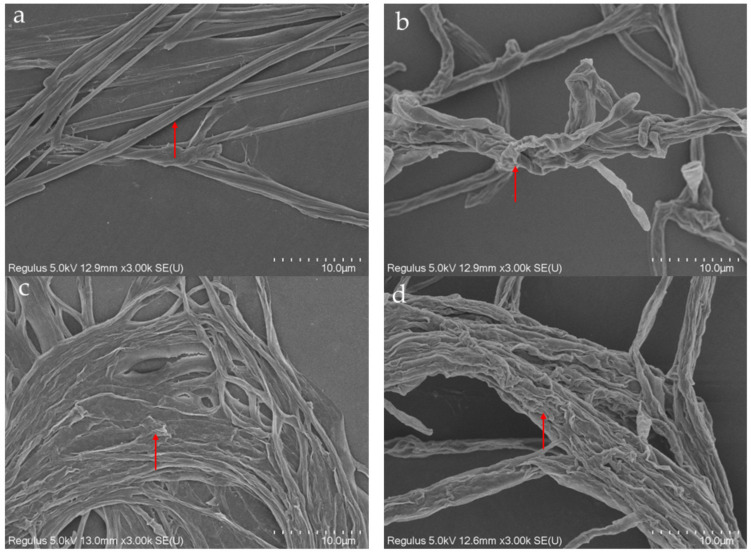
Scanning electron micrographs of mycelial morphology of *T. versicolor*: (**a**) blank control, ×3000; (**b**) treated with compound **3a** at 0.43 mg/L (EC_50_), ×3000; (**c**) treated with compound **4a** at 6.80 mg/L (EC_50_), ×3000; (**d**) treated with compound **5k** at 4.86 mg/L (EC_50_), ×3000. The red arrows point out the healthy mycelium and the damaged mycelium.

**Figure 5 jof-08-00762-f005:**
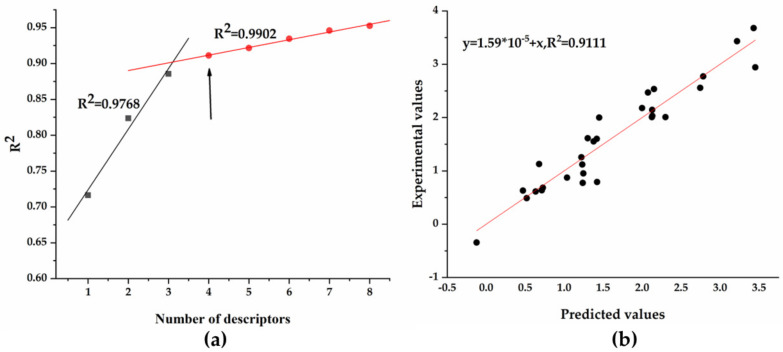
Confirmation of the number of descriptors and prediction of antifungal activity against *T. versicolor* using model constructed. (**a**) Application of the “breaking point” rule; (**b**) experimental and predicted log_50_ EC_50_ values.

**Figure 6 jof-08-00762-f006:**
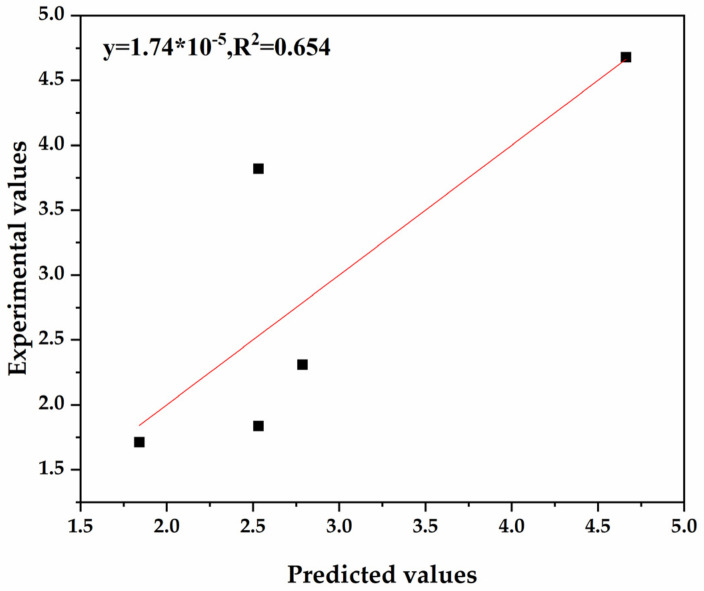
Comparison of the experimental values log_50_ EC_50_ and predicted values log_50_ EC_50_ based on external validation.

**Table 1 jof-08-00762-t001:** EC_50_ values of camphor-based derivatives against six fungi ^a^.

Compd.	EC_50_ (mg/L)					
	PN	FV	CG	SS	FO	TV
**1**	>1000	>1000	>1000	>1000	597.01	>1000
**2**	215.66	172.24	92.44	159.50	118.14	150.55
**3a**	25.08	40.18	12.85	17.09	19.30	0.43
**4a**	207.80	82.34	36.51	93.51	38.56	6.80
**4b**	253.14	88.85	34.94	109.93	15.89	99.70
**4c**	176.72	63.91	28.91	86.14	16.24	35.78
**4d**	158.81	69.10	30.62	101.23	19.97	7.67
**4e**	89.92	62.16	80.49	100.08	20.05	593.17
**4f**	>1000	>1000	74.00	>1000	734.52	361.20
**4g**	>1000	>1000	771.99	>1000	>1000	>1000
**4h**	79.78	34.39	65.66	69.47	29.85	101.99
**4i**	>1000	45.97	330.96	>1000	899.39	>1000
**4j**	101.81	39.68	26.90	54.04	38.55	68.47
**4k**	91.87	40.54	24.09	77.58	21.03	17.18
**4l**	178.16	58.83	59.23	120.53	20.80	204.45
**4m**	137.26	31.32	34.65	/	25.05	107.66
**4n**	174.28	66.43	35.03	127.07	16.33	342.56
**4o**	>1000	206.14	136.85	>1000	42.63	6.89
**4p**	341.97	131.69	33.22	162.35	38.88	18.86
**4q**	>1000	>1000	52.24	>1000	730.69	51.49
**4r**	73.59	645.97	80.55	101.04	26.50	>1000
**4s**	112.39	723.36	82.85	76.73	56.45	>1000
**5a**	>1000	125.46	371.75	>1000	>1000	4.42
**5b**	>1000	>1000	364.55	>1000	486.83	11.30
**5c**	>1000	816.15	639.90	>1000	>1000	7.40
**5d**	>1000	>1000	769.85	>1000	>1000	40.93
**5e**	>1000	>1000	522.47	>1000	>1000	139.57
**5f**	>1000	>1000	>1000	>1000	>1000	101.56
**5g**	>1000	>1000	652.17	>1000	>1000	7.85
**5h**	>1000	>1000	233.73	>1000	>1000	12.37
**5i**	>1000	>1000	>1000	>1000	/	/
**5j**	>1000	>1000	474.68	>1000	>1000	16.24
**5k**	>1000	>1000	489.08	>1000	>1000	4.86
**5l**	>1000	>1000	33.64	>1000	>1000	5.09
**5m**	>1000	>1000	>1000	>1000	/	294.86
**5n**	>1000	>1000	>1000	>1000	>1000	>1000
**5o**	>1000	>1000	634.710	>1000	/	>1000
tricyclazole	80.58	185.93	79.39	268.37	66.78	118.20
carbendazim	1.49	0.55	0.20	4.67	0.45	1.20

Note: ^a^ PN, *Phytophthora nicotianae*; FV, *Fusarium verticillioides*; CG, *Colletotrichum gloeosporioides*; SS, *Sphaeropsis sapinea*; FO, *Fusarium oxysporum*; TV, *Trametes versicolor*. “/” means no antifungal activity.

**Table 2 jof-08-00762-t002:** Details of the four-descriptor QSAR model.

Descriptor No.	Regression Coefficient (X)	Standard Error (ΔX)	*t*-Test	Descriptors
0	1.757	1.904	0.922	Intercept
1	0.416	0.052	8.067	NOF ^a^
2	0.903	0.1008	8.960	RMW ^b^
3	−11.104	2.862	−3.880	MAOEP ^c^
4	0.528	0.203	2.578	FPSA-2 ^d^

Note: ^a^ Number of F atoms; ^b^ Relative molecular weight; ^c^ Minimum atomic orbital electronic population; ^d^ FPSA-2 Fractional PPSA (PPSA-2/TMSA) [Quantum-Chemical PC].

**Table 3 jof-08-00762-t003:** Internal validation of QSAR model.

R^2^	R_loo_^2^	Training Set	N	R^2^ (Fit)	Test Set	N	R^2^ (Fit)
0.911	0.844	A + B	20	0.924	C	9	0.886
		B + C	19	0.886	A	10	0.831
		A + C	19	0.882	B	10	0.922
		Average	-	0.897	Average	-	0.880

## Data Availability

Not applicable.

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
