# Peer review of "Synthesis, Antifungal Activity, Cytotoxicity and QSAR Study of Camphor Derivatives"

_jof, 2022, doi:10.3390/jof8080762_

Round 1
Reviewer 1 Report
The work submitted by Dr. Shengliang Liao and coauthors describes the synthesis and antifungal essays for 34 camphor derivatives. Besides the interesting antifungal activities, which were evaluated against 6 phytopathogenic fungi , the authors complemented the study with the cytotoxic activity and morphologic and in silico studies.
The results were appropriately discussed, along with the chemical characterization for the synthesized compounds.
In my opinion, the manuscript is suitable to be accepted for publication in the Journal of Fungi after minor revisions pointed out as follows:
1) Line 212: the “N” in N-bromosuccinimide and the “p” in p-toluenesulfonic acid should be italicized.
2) In the supporting information, the “m/z“ also should be italicized.
3) In the supporting information, some of the given HRMS mass does not correspond to the expected accurate mass. For example, for compound 3a, the expected mass for [M+H]+ is 226.13xx and not 225,1378. Please, check the HRMS spectra in Figure S106.
This was also noticed for compound 4h. Please revise all given mass. I also suggest informing de calculated mass.
4) For some 13C NMR spectra, the baseline needs correction.
Author Response
Comment 1: The work submitted by Dr. Shengliang Liao and coauthors describes the synthesis and antifungal essays for 34 camphor derivatives. Besides the interesting antifungal activities, which were evaluated against 6 phytopathogenic fungi, the authors complemented the study with the cytotoxic activity and morphologic and in silico studies. The results were appropriately discussed, along with the chemical characterization for the synthesized compounds. In my opinion, the manuscript is suitable to be accepted for publication in the Journal of Fungi after minor revisions pointed out as follows:
Response: Thank you for the positive evaluation of our work and the kind suggestions.
Comment 2: Line 212: the “N” in N-bromosuccinimide and the “p” in p-toluenesulfonic acid should be italicized.
In the supporting information, the “m/z“ also should be italicized.
Response: We have made strict revisions to the existing formatting problems in the manuscript.
Comment 3: In the supporting information, some of the given HRMS mass does not correspond to the expected accurate mass. For example, for compound 3a, the expected mass for [M+H]+ is 226.13xx and not 225,1378. Please, check the HRMS spectra in Figure S106. This was also noticed for compound 4h. Please revise all given mass. I also suggest informing the calculated mass.
Response: We have checked the HRMS data and added the calculated mass.
Comment 4: For some 13C NMR spectra, the baseline needs correction.
Response: We have corrected the baseline of 13C NMR spectra.
All the modified contents in revised Manuscript and Supplementary materials have been marked in green.
Reviewer 2 Report
- Does Fusarium oxyporum is correct? and not Fusarium oxysporum?
- In the anti-fungal activity determination, why the authors did not used agar well diffusion technique instead of mycelium growth rate?
- lines 151 and 165: use superscript case for the power of ten (1*104 mg/L) and others through the manuscript.
Author Response
Comment 1: Does Fusarium oxyporum is correct? and not Fusarium oxysporum?
Response: Sorry for this clerical error. Fusarium oxysporum is correct, and we have corrected it.
Comment 2: In the anti-fungal activity determination, why the authors did not used agar well diffusion technique instead of mycelium growth rate?
Response: As far as we know, the agar well diffusion technique and the mycelial growth rate method can both be used to assess antifungal activity. The agar diffusion method is a very easy and useful method, it may be more suitable for fast growing aerobic bacteria and facultative anaerobic bacteria. In reality, however, all six of the fungi used in this experiment grew very slowly, it took more than a week to grow an entire petri dish. While, the mycelium growth rate method is one of the simple and common bioassay methods for fungicides, many scholars have used the mycelial growth rate method to assess the antifungal activity of compounds , here are some examples from the literature.
Mao, S.Y., et al. Pine Rosin as a Valuable Natural Resource in the Synthesis of Fungicide Candidates for Controlling Fusarium oxysporum on Cucumber. J. Agric. Food. Chem. 2021; 69: 6475−6484.
Yang, C., et al. "Semisynthesis and biological evaluation of some novel Mannich base derivatives derived from a natural lignan obovatol as potential antifungal agents - ScienceDirect." Bioorgan. Chem. 2019; 94: 103469.
Chen, Y.J., et al. "Design, Synthesis, and Antifungal Evaluation of Cryptolepine Derivatives against Phytopathogenic Fungi." J. Agric. Food. Chem. 2021,69.
W, M.L., et al. Design, Synthesis and Antifungal/Anti‐Oomycete Activity of Pyrazolyl Oxime Ethers as Novel Potential Succinate Dehydrogenase Inhibitors. Pest Manag. Sci. 2020.
Comment 3: lines 151 and 165: use superscript case for the power of ten (1*104 mg/L) and others through the manuscript.
Response: We quite appreciate your kind suggestions; we have carefully revised the manuscript according to your comments.
All the modified contents in revised Manuscript and Supplementary materials have been marked in blue.
Reviewer 3 Report
The manuscript entitled “Synthesis, Antifungal activity of camphor-based derivatives against phytopathogenic fungi and Cytotoxicity and QSAR study” and coded as jof-1794191 aimed at providing a theoretical basis for the high-value utilization of camphor and could be helpful for the development of novel potential fungicides against phytopathogenic fungi. In general, the authors have done extensive work for their publication. However, the article should be improved before publishing. Some comments are appended below.
Lines 27-28: It is unnecessary to place precise EC50 values since the fungal species are not described in the abstract; it is suggested to place only the range.
Lines 58-67: Synthesize the text with the examples without citing them in detail. It reads redundant due to the similarity of the works.
Line 78: Are not antifungal agents expected to have fungicidal action?
Lines 104-108: You tested against 6 phytopathogenic fungi; how were these fungi identified by taxonomy using dichotomous keys or molecular methods?
As they can be sure that the fungi are phytopathogenic, they could be saprobes. Did they demonstrate its pathogenicity in any previous study?
Line 106: The current name according to Coriolus versicolor is Trametes versicolor. Change please. http://www.speciesfungorum.org/Names/SynSpecies.asp?RecordID=281625
Cytotoxicity studies, SEM and QSAR, are performed with C. versicolor; however, this fungus has mainly saprobic habits rather than phytopathogenic. Where was this strain isolated from? Due to its saprobic relevance, the greater susceptibility would make sense since this fungus has probably not been exposed to fungicidal compounds compared to other species that are frequently reported as phytopathogenic.
In the electronic microscopy, why did they not perform the analysis of the controls (commercial fungicides)?
In Figure 1, compound 5l is repeated.
In Figure 2, place the results obtained by Tricyclazole and Carbendazim. In addition, they must improve the quality of the image, such as the one shown in their article by Zhang et al. 2021.
Were the antifungal assays done in triplicate? If so, place the standard deviations in Table 1. If they did not do it in triplicate, explain why and how we could verify the reproducibility of the assay.
Line 232. Place the lowest EC50 of Fusarium oxysporum; it should be 15.89
It was not possible to review the supplementary material; it is very extensive and complex. They only place Supplementary materials in the text; they do not refer to a Table or Figure, making it difficult for readers.
Line 264: What do you mean by “both’’
Lines 276-278: Cite references that support this assumption.
Provide further discussion in the evaluation of the cytotoxicity section.
Fig. 4, Add at least one pair of SEM images per treatment in supplementary material, showing raw data. This will prove that the submitted images were not selected.
Fig. 4. Point out possible damage to the images. The material as processed shows no apparent damage. What is shown appears to be damage from sample processing (cell collapse due to electron emission).
The first descriptor in the QSAR study is the number of fluorine atoms. However, not all molecules have fluorine atoms.
I understand that the antifungal mechanisms were not investigated, but based on previous studies with camphor (Zhang et al. 2021), suggest the possible mechanisms of antifungal action.
Author Response
Comment 1: Lines 27-28: It is unnecessary to place precise EC50 values since the fungal species are not described in the abstract; it is suggested to place only the range.
Response: We appreciate your suggestion, and we have modified the expression in the revised manuscript.
Comment 2: Lines 58-67: Synthesize the text with the examples without citing them in detail. It reads redundant due to the similarity of the works.
Response: We appreciate your suggestion, and we have modified the expression in the revised manuscript.
Comment 3: Line 78: Are not antifungal agents expected to have fungicidal action?
Response: We quite appreciate your thoughtful comment, and we have removed the duplicate words in the manuscript.
Comment 4: Lines 104-108: You tested against 6 phytopathogenic fungi; how were these fungi identified by taxonomy using dichotomous keys or molecular methods? As they can be sure that the fungi are phytopathogenic, they could be saprobes. Did they demonstrate its pathogenicity in any previous study?
Response: Thank you for the questions. We have carefully checked the six fungi used in this study, and confirmed that five phytopathogenic fungi, namely Phytophthora nicotianae (P. nicotianae), Fusarium verticillioides (F. verticillioides), Colletotrichum gloeosporioides (C. gloeosporioides), Sphaeropsis sapinea (S. sapinea) and Fusarium oxysporum (F. oxysporum) were isolated and identified by the plant pathology laboratory at the College of Agriculture, Jiangxi Agricultural University; and another wood-rotting fungus Trametes versicolor (T. versicolor, cfcc5336) was purchased from China Forestry Culture Collection Center.
Comment 5: Line 106: The current name according to Coriolus versicolor is Trametes versicolor. Change please.http://www.speciesfungorum.org/Names/SynSpecies.asp?RecordID=281625.
Response: We appreciate your suggestion. We have checked the name of this fungus, and corrected it.
Comment 6: Cytotoxicity studies, SEM and QSAR, are performed with C. versicolor; however, this fungus has mainly saprobic habits rather than phytopathogenic. Where was this strain isolated from? Due to its saprobic relevance, the greater susceptibility would make sense since this fungus has probably not been exposed to fungicidal compounds compared to other species that are frequently reported as phytopathogenic.
Response: We quite appreciate your thoughtful comments! Trametes versicolor (T. versicolor, cfcc5336) was purchased from China Forestry Culture Collection Center. Trametes versicolor is a wood-rotting fungus. Meanwhile, during the mushroom cultivation, Trametes versicolor is often treated as a harmful "miscellaneous fungus". Therefore, it is of great significance to develop some new antifungal agents against this fungus.
Comment 6: In the electronic microscopy, why did they not perform the analysis of the controls (commercial fungicides)?
Response: We focused on whether the synthesized compound had stronger antifungal activity than the raw material, so only several derivatives with high antifungal activity were analyzed by scanning electron microscopy. However, the evaluation of positive controls (antifungal agents) was neglected. This is a flaw in our experimental design. We will improve the experiment design in the following work. We quite appreciate your helpful comments!
Comment 7: In Figure 1, compound 5l is repeated.
Response: Thanks, we have revised the error.
Comment 8: In Figure 2, place the results obtained by Tricyclazole and Carbendazim. In addition, they must improve the quality of the image, such as the one shown in their article by Zhang et al. 2021.
Response: Due to the large amount of data, it is difficult to put the results of derivatives and positive controls on the same Figure. However, we will add the photos of positive controls in the supplementary material. For the original photos of petri dishes were taken in an automatic colony counter when the UV lamp was turn on, the background of these photos is not clear enough. Now, the petri dishes have been cleaned, so there's no way to re-photograph them. In the future work, we will try our best to improve the qualityof the image. We quite appreciate your helpful comments!
Comment 9: Were the antifungal assays done in triplicate? If so, place the standard deviations in Table 1. If they did not do it in triplicate, explain why and how we could verify the reproducibility of the assay.
Response: The antifungal assays were conducted in triplicate. We measured the diameters of fungus cakes in three parallel experiments by crossover method and calculated the inhibition rate respectively. The mean value and standard deviation of the inhibition rates obtained in the three parallel experiments were calculated. EC50 was calculated using the average inhibition rates at different concentrations. We have added the standard deviations of the inhibition rate in the supplementary material. We quite appreciate your helpful comments!
Comment 10: Line 232. Place the lowest EC50 of Fusarium oxysporum; it should be 15.89.
Response: Sorry for this clerical error. We have revised it, thank you.
Comment 11: It was not possible to review the supplementary material; it is very extensive and complex. They only place Supplementary materials in the text; they do not refer to a Table or Figure, making it difficult for readers.
Response: We apologize for any confusion caused by the complexity of the Supplementary materials. Due to the large amount of data involved in this study, such as derivative synthesis, bioactivity evaluation and QSAR, some data and pictures can only be placed in the Supplementary materials. The purpose of presenting these data and pictures is to give readers a more comprehensive understanding of our work.
Comment 11: Line 264: What do you mean by “both’’?
Response: We have removed the word “both” in the revised manuscript.
Comment 12: Lines 276-278: Cite references that support this assumption.
Response: We quite appreciate your helpful comments! We have cited some references.
Comment 13: Provide further discussion in the evaluation of the cytotoxicity section.
Response: We quite appreciate your helpful comments! We have added some further discussion in the evaluation of the cytotoxicity section.
Comment 14: Fig. 4, Add at least one pair of SEM images per treatment in supplementary material, showing raw data. This will prove that the submitted images were not selected.
Response: We have added one pair of SEM images per treatment in supplementary material. We quite appreciate your thoughtful comments!
Comment 15: Fig. 4. Point out possible damage to the images. The material as processed shows no apparent damage. What is shown appears to be damage from sample processing (cell collapse due to electron emission).
Response: We have recreated Figure 4, which the red arrows were used to point out possible damage. We quite appreciate your helpful comments!
Comment 16: The first descriptor in the QSAR study is the number of fluorine atoms. However, not all molecules have fluorine atoms.
Response: In QSAR research, structure parameterization is a very important process. In this process, each compound can be parameterized and hundreds of descriptors can be obtained. All descriptors have specific numeric values. In the case of the descriptor "the number of fluorine atoms", if a compound does not contain fluorine, the numeric value of this descriptor for that compound is zero; if a compound contains two fluorine, the numeric value of this descriptor for that compound is two, and so on.
Comment 17: I understand that the antifungal mechanisms were not investigated, but based on previous studies with camphor (Zhang et al. 2021), suggest the possible mechanisms of antifungal action.
Response: We quite appreciate your thoughtful comments! In the section “3.4. Effect on the Mycelium Morphology of T. versicolor”, we preliminarily speculated the mechanism of action of camphor-based derivatives(line 322-325 in revised manuscript: Combined with previous research findings [41-42], it can be speculated that the fungal cell walls and cell membranes of T. versicolor were potential targets of the camphor-based derivatives, but the mechanism of action of camphor-based derivatives needs to be further studied in more detail.). At the same time, we are conducting research on the antifungal mechanism of action of camphor-based derivatives, and detailed results will be reported in the future.
All the modified contents in revised Manuscript and Supplementary materials have been marked in red.
Reviewer 4 Report
The novelty of this study is not clearly mentioned in the introduction section.
The introduction must be formulated and recent references added
Antibacterial activity and virtual screening by molecular docking of lycorine from Pancratium foetidum Pom (Moroccan endemic Amaryllidaceae)
Microbial Pathogenesis, Volume 115, February 2018, Pages 138-145
H. Bendaif, A. Melhaoui, M. Ramdani, H. Elmsellem, Y. El Ouadi
Antibacterial, antifungal and antioxidant activity of total polyphenols of Withania frutescens.L
Bioorganic Chemistry, Volume 93, December 2019, Article 103337
Abdelfattah El Moussaoui, Fatima Zahra Jawhari, Ahmed M. Almehdi, Hicham Elmsellem, Amina Bari
. The novelty of the work be established.
. Give information about all the chemicals and apparatus used throughout the study.
. Abstract and conclusions are poor. Highlights some core findings of the investigation.
In order to sustain the conclusions other techniques should be used
Author Response
Comment 1: The novelty of this study is not clearly mentioned in the introduction section. The introduction must be formulated and recent references added.
Antibacterial activity and virtual screening by molecular docking of lycorine from Pancratium foetidum Pom (Moroccan endemic Amaryllidaceae) Microbial Pathogenesis, Volume 115, February 2018, Pages 138-145 H. Bendaif, A. Melhaoui, M. Ramdani, H. Elmsellem, Y. El Ouadi.
Antibacterial, antifungal and antioxidant activity of total polyphenols of Withania frutescens.L Bioorganic Chemistry, Volume 93, December 2019, Article 103337. Abdelfattah El Moussaoui, Fatima Zahra Jawhari, Ahmed M. Almehdi, Hicham Elmsellem, Amina Bari.
Response: Thanks, these references have been added.
Comment 2: The novelty of the work be established.
Response: We quite appreciate your positive comment!
Comment 3: Give information about all the chemicals and apparatus used throughout the study.
Response: We quite appreciate your helpful comments! We have given information in the section “2.1. Materials and Equipment”.
Comment 4: Abstract and conclusions are poor. Highlights some core findings of the investigation. In order to sustain the conclusions other techniques should be used.
Response: We quite appreciate your insightful comments!
All the modified contents in revised Manuscript and Supplementary materials have been marked in orange.
Round 2
Reviewer 3 Report
1. “miscellaneous fungi”. First, place in the singular. Second, what do they mean by miscellaneous, in fungal taxonomy refers to a list of species from different substrates. Here, they specify the phytopathogens and Trametes versicolor. Trametes versicolor rots the wood, however this is a process of saprobic nutrition that is part of the recycling of nutrients in the ecosystem. I suggest placing a couple of lines justifying the tests of camphor against this fungus, e.g., that T. versicolor is a common polypore species found throughout the world, thanks to its ease of growth under different types of environments, could suggest that camphor compounds can be applied to any pathogenic fungus.
2. Lines 28: Soften the sentence. They did not perform minimum fungicidal concentration to determine that the compounds kill the fungus. By SEM, death cannot be inferred since the cell could continue to be viable after the tests; the camphor derivatives only inhibited it.
3. To comment 8 of review 1. Add on line 243 to which figure of supplementary material the photographs of the controls refer.
4. Concerning comment 14 of revision 1, Add-in section 3.4, the figure in supplementary materials to which they refer to the SEM images.
5. They must place throughout the manuscript which Tables or Figures of supplementary material they refer to.
Author Response
omment 1: “miscellaneous fungi”. First, place in the singular. Second, what do they mean by miscellaneous, in fungal taxonomy refers to a list of species from different substrates. Here, they specify the phytopathogens and Trametes versicolor. Trametes versicolor rots the wood, however this is a process of saprobic nutrition that is part of the recycling of nutrients in the ecosystem. I suggest placing a couple of lines justifying the tests of camphor against this fungus, e.g., that T. versicolor is a common polypore species found throughout the world, thanks to its ease of growth under different types of environments, could suggest that camphor compounds can be applied to any pathogenic fungus.
Response: We quite appreciate your suggestion, and we have modified the expression in the revised manuscript. Meanwhile, we have placed a couple of lines justifying the tests of camphor-based derivatives against this fungus at page 6 of the revised manuscript.
Comment 2: Lines 28: Soften the sentence. They did not perform minimum fungicidal concentration to determine that the compounds kill the fungus. By SEM, death cannot be inferred since the cell could continue to be viable after the tests; the camphor derivatives only inhibited it.
Response: We quite appreciate your suggestion, and we have modified the expression in the revised manuscript.
Comment 3: To comment 8 of review 1. Add on line 243 to which figure of supplementary material the photographs of the controls refer.
Response: We quite appreciate your suggestion, and we have added the relevant information in the revised manuscript.
Comment 4: Concerning comment 14 of revision 1, Add-in section 3.4, the figure in supplementary materials to which they refer to the SEM images.
Response: We quite appreciate your suggestion, and we have added the relevant information in the revised manuscript.
Comment 5: They must place throughout the manuscript which Tables or Figures of supplementary material they refer to.
Response: We appreciate your suggestion, and we have checked in full and added the relevant information in the revised manuscript.
This manuscript is a resubmission of an earlier submission. The following is a list of the peer review reports and author responses from that submission.